# Hybrid Optimization Algorithm Based on Double Particle Swarm in 3D NoC Mapping

**DOI:** 10.3390/mi14030628

**Published:** 2023-03-09

**Authors:** Juan Fang, Huayi Cai, Xin Lv

**Affiliations:** Faculty of Information Technology, Beijing University of Technology, Beijing 100124, China

**Keywords:** 3D NoC, particle swarm optimization, neighborhood search, gene cross-mutation, high performance computing

## Abstract

Increasing the number of cores on a chip is one way to solve the bottleneck of exponential growth but an excessive number of cores can lead to problems such as communication blockage and overheating of the chip. Currently, networks-on-chip (NoC) can offer an effective solution to the problem of the communication bottleneck within the chip. With current advancements in IC manufacturing technology, chips can now be 3D-stacked in order to increase chip throughput as well as reduce power consumption while reducing the area of the chip. Automating the mapping of applications into 3D NoC topologies is an important new direction for research in the field of 3D NoC. In this paper, a 3D NoC partitioning algorithm is proposed, which can delineate the 3D NoC region to be mapped. Additionally, a double particle swarm optimization (DPSO) based heuristic algorithm is proposed, which can integrate the characteristics of neighborhood search and genetic algorithms, and thus solve the problem of a particle swarm algorithm falling into local optimal solutions. It is experimentally demonstrated that this DPSO-based hybrid optimization algorithm has a higher throughput rate and lower energy loss than the traditional heuristic algorithm.

## 1. Introduction

With the advent of the digital era, the improvement of arithmetic power has become an increasingly critical issue in the development of computers today. Integrated circuits have evolved rapidly in just a few decades. Hundreds of millions of transistors can now be integrated into a single chip, allowing for a great increase in computer processing power. However, the external I/O of the chip has become the bottleneck of the chip’s computing power, and so to solve this problem, system-on-chip (SoC) technology was born. With an increase in the number of cores in a single chip, inter-core communication becomes a key issue to be considered in SoC design, which uses a bus-based architecture wherein multiple cores and internal storage devices are connected to the same bus and exchange data through the bus. However, with the increase in the number of IP cores, the data exchange load on the bus increases. Therefore, network-on-chip (NoC) architecture is proposed in order to relieve the bus load. However, as the number of IP cores continues to increase, there are some issues to be faced with 2-Dimensional (2D) NoCs, such as chip area, performance, bandwidth, and power consumption. The 3-Dimensional (3D) NoC can effectively reduce the size of system interconnects for large-scale SoCs, eliminating the limitations of 2D NoCs in terms of chip area and performance. A 3D NoC can vertically accumulate multiple 2D NoCs and package them into a single chip, increasing the data transfer in the vertical dimension. The longitudinal interconnection of chips relies on through-silicon-via (TSV) technology. The technique of stacking 3D chips using TSV technology is a key to overcoming the problem of the chip performance bottleneck [1]. When the number of nodes is the same, 3D NoC has a smaller area, lower delay, and lower power consumption than 2D NoC, and can disperse data transmission in more links, therefore improving overall system performance and power consumption.

However, oversized networks can result in reduced heat dissipation and heat buildup. This can increase the failure rate of the IP core, with approximately every 10 degree increase in temperature causing a 5% increase in system data transmission latency [2]. In addition, the instantaneous increase in temperature can result in severe system failures, and prolonged overheating can result in the deterioration of the chip’s packaging materials and integrated circuits. This can result in increased cooling costs, reduced reliability, and signal integrity issues [3,4,5]. At present, overheating problems are a major issue in the field of IC design, which has led to the emergence of various chip cooling techniques. In many cases, these cooling techniques, such as dark silicon [6,7], can limit the performance of the chip, by cutting off or reducing the power supply to some of the IP cores in order to avoid failures that occur due to high temperatures. To ensure heat dissipation and uniform heat distribution in 3D NoC networks, a heat-sensitive application mapping strategy is often used. 3D NoC research is usually divided into three parts: infrastructure, routing algorithm, and task mapping. Task mapping is a key area of 3D NoC research and has a direct impact on the power consumption, latency, and other performance metrics of communication. There have been many studies dedicated to the design of mapping algorithms for 3D NoC from different perspectives. Fang et al. [8] proposed the KL_GA algorithm, which uses the Kernighan–Lin algorithm to generate a mapping result, and uses a genetic algorithm to avoid the occurrence of prematureness and avoid falling into a local optimum solution. Refs. [9,10] used the particle swarm algorithm to solve the mapping problem of NoCs. Simulated annealing algorithms, etc., have also been used. Mohammad et al. [11] used the boundary mapping algorithm (BMA), which reduces the priority of low-weight edges in the task graph by proposing a method based on cellular learning automata. Li et al. [12] proposed a runtime mapping algorithm for TSV technology, which can divide a specific rectangular area in the NoC for the application based on the number of occupied vertical links and the distance of the heat sink, and optimize communication performance by reducing the peak temperature, and then map the subtasks to the NoC resource nodes after determining the area. Yu et al. [13] proposed a genetic algorithm based on a mapping algorithm which uses a greedy algorithm to generate the initial population and adds a simulated annealing algorithm in the cross-variance phase of the genetic algorithm in order to avoid getting trapped in a locally optimal solution. Aravindhan et al. [14] proposed a knowledge-based memetic algorithm (KBMA). A memetic algorithm is a combination of the evolutionary algorithm (EA) and local search (LS) algorithm. The EA algorithm is good for solution space exploration, while the LS algorithm is better at finding optimal solutions locally and uses power consumption, area, and delay as evaluation metrics. Nalci et al. [15] proposed an integer linear programming (ILP) formulation and a novel heuristic algorithm called CastNet3D, with the goal of energy minimization. The algorithm utilizes as many vertical links as possible to communicate between NoC resource nodes

Hybrid optimization algorithms are usually applied in complex nonlinear systems. Zhang et al. [16] proposed an orientation function-based prescribed performance control (PPC). Ding et al. [17] indicated that the conventional linear and nonlinear models would fail for long-term prediction and an evolutionary algorithm is utilized to obtain the optimal model for long-term prediction. Zhang et al. [18] proposed an image enhancement algorithm; by combining the rough set theory with a Gaussian mixture model, an algorithm with higher immunity is proposed.

The task mapping is divided into two stages: the mapping of task nodes to IP cores and the mapping of IP cores to NoC maps. In this paper, based on the analysis of the network scale of 3D NoC and the existing NoC benchmark, we propose an IP core to NoC mapping by region, in order to achieve an even distribution of task load while taking into account the heat dissipation efficiency, in order to reduce the chip temperature as much as possible and improve the processing capacity of the chip. The problem of IP core to NoC node mapping is an NP-hard problem. In this paper, we propose a particle swarm-based thermal-aware mapping algorithm in order to find the optimum in the solution space.

In this work, in order to prevent the PSO algorithm from prematurely falling into the local optimal solution, this paper proposes a double particle swarm optimization algorithm. One of the particle populations introduces cross-mutation to improve global search capabilities. Another particle population incorporates neighborhood search to improve local search capabilities. When the two populations update their vectors of positions to the global optimal solution and the local optimal solution, they also update to the optimal solution position of the other population. A dynamic inertia factor is proposed to control the convergence speed.

## 2. System and Computation Model

In the field of on-chip networking, routing, scheduling, and task mapping are all hot topics. Each application can be divided into several independent subroutines that can be assigned in different IP cores at runtime to exchange data over links and thus coordinate the overall output of the final result. Thus, each application can be represented by a directed acyclic graph (DAG). Each node of the graph represents a subroutine, the edges represent the data transfer relationships of the subroutines, and the weights of the edges represent the amount of data communication of the connected subroutines. Each resource node can execute any subroutine (a node in the DAG) of the application. Determining how to assign the sub-nodes of the DAG graph to the resource node in the NoC is the problem addressed by task mapping. In this paper, the mapping problem is solved by the optimization of power consumption, latency and the three aspects of thermal energy.

### 2.1. Task Graph

The task graph used is usually a weighted DAG. In a task graph $TG\left(V,K\right)$ the vertex $v_{i}\in V$, which represents a subtask in the application. The directed edge $k_{m,n}\in K$, which contains both direction and weight information (i.e., the data transmission direction is from the $e_{m}$ node to the $e_{n}$ node). The weight indicates the amount of data communicated in these two subtasks. If no communication relationship is formed between the $e_{m}$ node and the $e_{n}$ node, then $k_{m,n}=0$. Equation (1) represents the communication relatiohip matrix.
(1)$$K=\left[k_{i,j}\right]=\left[\begin{array}{ccc} 0 & \cdots & k_{0,n-1}\\ \vdots & \ddots & \vdots \\ k_{n-1,0} & \cdots & 0 \end{array}\right]$$

### 2.2. 3D NoC Topology

In this paper, the NoC mapping algorithm under a 3D mesh structure is discussed. The 3D NoC mapping system consists of resource nodes, routings, physical links and TSV links, as shown in Figure 1. The NoC topology graph is defined as NoC(U,L). Where the vertex $u_{i}\in U$ denotes the resource node in NoC, $l_{i}\in L$ represents the pathway of the communication link in NoC. $N_{U}=X\times Y\times Z$ and where $N_{U}$ is the number of resource nodes in the U set, X×Y represents the number of resource nodes in each layer; Z represents the number of layers in this 3D NoC system. The size of different layers in 3D NoC is the same. The vertical connection of nodes through TSV technology. For example, the size of 3D NoC shown in Figure 1 is 3×3×3; through the coordinates, from (1,1,1) to (3,3,3), the unique resource node can be identified.

### 2.3. Communication Power Consumption Model

Communication power consumption is an important metric used for describing the performance of on-chip networks. It refers to the communication power consumption of each node in the NoC system and the power consumption generated by the data in the link. Its size is related to the distance the data slices are transmitted between nodes, the type of transmission path (Link or TSV), and the amount of communication transmitted. The model used in this paper is a generic power calculation model for a 3D-mesh structured NoC, i.e., the energy consumption required for 1-bit data transmission over the NoC, as shown in Equation (2).
(2)$$E_{bit}=E_{Sbit}+E_{Bbit}+E_{Lbit}+E_{Wbit}$$

$E_{Bbit}$ and $E_{Wbit}$ represent the energy consumption of 1-bit data on the cache and internal link lines, respectively. This energy consumption is relatively small and usually neglected; $E_{Sbit}$ and $E_{Lbit}$ represent the energy consumed by 1-bit data on the router and adjacent routing node links, respectively. Thus, Equation (1) can be simplified to Equation (3).
(3)$$E_{bit}=E_{Sbit}+E_{Lbit}$$

Therefore, the energy consumed by 1-bit data from the $e_{m}$ node to the $e_{n}$, $E_{bit}^{mn}$ node is shown in Equation (4).
(4)$$\left\{\begin{array}{c} E_{bit}^{mn}=E_{Sbit}^{mn}+E_{Lbit}^{mn}\\ E_{Sbit}^{mn}=N_{dimx}\times E_{Sbit}^{\prime }+N_{dimy}\times E_{Sbit}^{\prime }+N_{dimz}\times E_{Sbit}^{\prime }+\\ E_{Lbit}^{mn}=N_{dimx}\times E_{Lbit}^{\prime }+N_{dimy}\times E_{Lbit}^{\prime }+N_{dimz}\times \mu E_{Lbit}^{\prime } \end{array}\right.E_{Sbit}^{\prime }$$

$E_{Sbit}^{\prime }$ and $E_{Lbit}^{\prime }$ represent the energy consumed by 1-bit of data in a router and an inter-route link. $N_{dimx}$, $N_{dimy}$ and $N_{dimz}$ represent the number of links that pass through the X-dimension, Y-dimension, and Z-dimension from node $e_{m}$ to node $e_{n}$, provided that the shortest possible routing algorithm is adopted. μ represents the energy consumption of data in transmission in TSV and the ratio of energy consumption in horizontal links (TSV consumes only 7.5% of the communication energy compared to normal horizontal links [4]).

Assuming that the size of the data volume from the $e_{m}$ node to the $e_{n}$ node is $B_{mn}$, the total power consumption $E_{total}^{mn}$ from the $e_{m}$ node to the $e_{n}$ node can be derived according to Equation (4), as shown in Equation (5).
(5)$$E_{total}^{mn}=B_{mn}\times E_{bit}^{mn}$$

The total communication energy consumption of the NoC chip is obtained by adding up the communication consumption of all resource nodes and transmission links in the NoC. This is shown in Equation (6).
(6)$$E_{total}=\sum_{m,n\in NoC}E_{total}^{mn}$$

### 2.4. Delay Model

The delay of the NoC network is an important metric for describing the on-chip network performance. It refers to the time taken to perform the task of each node in the NoC system, the delay generated by the data in the link and the delay of data transmission in the router. The Delay in the NoC system is shown in Equation (7).
(7)$$D_{bit}=\sum D_{link}+\sum D_{core}+\sum D_{router}$$
where $D_{link}$ represents the delay in the data on the transmission link, $D_{core}$ represents the delay in the data in the resource node, and $D_{router}$ represents the delay in the data in the router, as shown in Equation (8).
(8)$$\left\{\begin{array}{c} D_{link}^{m,n}=\beta_{link}\times d_{horizon}^{m,n}\\ D_{core}^{n}=\beta_{core}\\ D_{router}^{m,n}=\beta_{router}\times \left(d^{m,n}+1\right) \end{array}\right.$$

$\beta_{link},\;\beta_{core}$ and $\beta_{router}$ are constants that represent the transmission delay of 1-bit data in the link, resource node and router, and $d_{horizon}^{m,n}$ is the shortest path length from the $e_{m}$ node to the $e_{n}$ node in the horizontal direction, because the data transmission path in the vertical direction is achieved by TSV technology and its delay is negligible compared to the normal technology. $d^{m,n}$ is the shortest path length from the $e_{m}$ node to the $e_{n}$ node.

Assuming that the size of the data volume from the $e_{m}$ node to the $e_{n}$ node is $B_{mn}$, the total communication delay $D_{total}^{mn}$ from the $e_{m}$ node to the $e_{n}$ node can be derived according to Equations (7) and (8), as shown in Equation (9).
(9)$$D_{total}^{m,n}=B_{mn}\times D_{bit}$$

The total communication delay on the NoC chip is obtained by calculating the total communication consumption of all resource nodes and transmission links in the NoC. This is shown in Equation (10).
(10)$$D_{total}=\sum_{m,n\in NoC}D_{total}^{mn}$$

### 2.5. Thermal-Aware Model

In this subsection, we formulate the computation offloading problem in order to optimize the overhead of applications generated by $UE_{i}$.

In Section 1, the hazards of high chip temperatures are mentioned. Therefore, this paper introduces the processor’s thermal sensing model into consideration for algorithm optimization. The temperature of a single resource node depends on the energy consumption of the resource node, the temperature of neighboring nodes, and the distance between the processor and the heatsink [19]. Hotspot [20] is a model that can calculate the thermal energy efficiency of the resource node in a chip. It takes into account the heat dissipation within the resource node and the heat transfer between adjacent processing units, from which the temperature of the resource node is calculated. For adjacent nodes, Hotspot calculates the thermal resistance $R_{m,n}$ from node $e_{m}$ to node $e_{n}$ by calculating the length and ratio of the adjacent edges between two resource nodes. This is shown in Equation (11).
(11)$$R_{m,n}=\Delta T_{m,n}/E^{n}$$
where $\Delta T_{m,n}$ represents the temperature at which the node $e_{m}$ rises, and $E^{n}$ represents the power consumption of the (previously mentioned) $e_{n}$ node.

It is assumed that all resource nodes in a unified NoC system interact with each other, so the temperature of each resource node can be calculated by Equation (12).
(12)$$T_{i}=\underset{j\in NoC}{\sum }R_{i,j}\times E^{j},\;i\in NoC$$

Assuming that the number of resource nodes in NoC is N, the peak temperature $T_{peak}$ in the model is shown in Equation (13).
(13)$$T_{peak}=\max\left\{T_{i}\right\},\;i\in NoC$$

The above calculation process is entirely integrated into the Hotspot thermal model. This is the most commonly used thermal sensing model in NoC computing systems. In this paper, Hotspot is used to quantify energy consumption.

## 3. Proposed Algorithm

In this paper, the mapping algorithm is divided into two phases: division of the 3D NoC operation region, and the mapping of tasks to NoC resource nodes. In 2D NoC, because the chip area is limited, the number of integrated resource nodes is usually less than the number of subtasks of the application, so there can be multiple subtasks mapped to the same resource node. In contrast, 3D NoC uses chip stacking technology to increase the layers of the chip through TSV, and this increases the number of resource nodes so that the number of resource nodes is larger than the number of application subtasks. Part of the 3D NoC area is delineated to map subtasks, allowing subtasks to transfer data with the shortest possible communication distance, which usually results in the lowest power consumption and latency.

### 3.1. Regional Division

In 3D NoC, the expansion of the network dimension makes the number of network computing nodes grow exponentially, and the performance of 3D NoC cannot be compared with traditional NoC, so concentrating the mapped nodes when running a single application can effectively avoid the delay and power consumption caused by long pathways during data transmission, and thus avoid the heat buildup generated by power consumption. In order to address these characteristics, we seek to improve the convergence speed of the particle swarm algorithm, by performing region partitioning on 3D NoC in advance based on the thermal resistance theorem and communication delay. The sub-nodes of the application are run in the partitioned region in order to achieve the lowest possible communication delay and heat buildup. Considering that the pathway distance directly affects the communication delay, we prioritize the node with the shortest pathway length in our NoC subgraph. If there are multiple nodes with the same pathway distance to the NoC subgraph, we calculate the impact of each of these nodes on the heat propagation of other nodes in the NoC subgraph according to the thermal resistance formula, and select the node corresponding to the smallest of them, then add it to the NoC subgraph. Because the TSV technique has a low delay on the vertical link, it is preferable to select the computational nodes in the vertical link. Figure 2 shows an example of regional division.

First, we define the number of CV nodes of the TG graph to determine the minimum number of computational nodes for the partitioned NoC region. We also define a hypothetical required energy consumption value ϵt in each node to calculate the hypothetical temperature increase value of the node based on the thermal resistance formula in order to predict the performance of the partitioned NoC region in terms of communication delay and temperature when executing the application.

The process of NoC regional division algorithm is shown in Algorithm 1.

The temperature metrics of each node are calculated based on the energy consumption value
ϵ and added to the temperature queue
TL (lines 1–3), then arranged in ascending order. The smallest temperature node is found as the initial division node and added to the resource node group VG (lines 4–5).

2.Iterate through all nodes connected by TSV technology in the
VG
group and add them to the queue
L (lines 7–9). If there are no more vertically adjacent (connected by TSV) nodes, the horizontally adjacent (connected by common link) nodes of all nodes in the
VG group are added to the queue L.

3.The sum of the horizontal Manhattan distances from the nodes in the queue
L to the nodes in the
VG are calculated (lines 11–12). The values are then added to the queue and sorted, and the minimum value is selected (lines 23–29). If multiple nodes have the same horizontal distance to the VG, their effect on the thermal resistance value in the VG is calculated (lines 13–15). The minimum operator node is then selected and added to the
VG group (lines 16–22).

4.If the number of nodes in the
VG is less than the number of nodes CV in the
TG graph, steps 2 to 4 are repeated.


**Algorithm 1**: NoC Regional Division Algorithm
**Inputs:**
$NoC\left(U,L\right)$: NoC topology graph
X,Y,Z: The number of NoC resource nodes in the three dimensions
$u\left(x,y,z\right)$: Resource node with coordinates x, y, z
$TG\left(V,K\right)$: Application task graph of the application, where the number of the task graph is CV${є}_{t}$: Assumed power consumption of each alternative node

**Ensure:**
VG: Group of resource node

Optimal offloading policy *A*, the total overhead *C*
1: **For**
*u* є 𝑈 **do**2:   temperature list:
$TL\leftarrow \sum_{u\in NoC}R_{i,j}\times \epsilon_{t}$
3: **End for**4: Get the coordinate of the
$v\left(x_{t},y_{t},z_{t}\right)\leftarrow \min\left(TL\right)$
5:
$VG\leftarrow v\left(x_{t},y_{t},z_{t}\right)$. 6: **While** $num\left(VG\right)<C_{V}-1$ **do**
7:   **If**
*Z_t_* + 1 ≤ *Z*
**then**
8:     
$VG\leftarrow v\left(x_{t},y_{t},z_{t}\right)$
9:      $z_{t}\leftarrow z_{t}+1$
10:  **Else**11:    $dist_{x}\leftarrow \underset{v\in VG}{\sum }\left|x_{t}+1-X\left(v\right)\right|+\left|y_{t}-Y\left(v\right)\right|$
12:    $dist_{y}\leftarrow \sum_{v\in VG}\left|X_{t}-X\left(v\right)\right|+\left|y_{t}+1-Y\left(v\right)\right|$
13:    **If** $dist_{x}==dist_{y}$
**then**14:       $T_{x}\leftarrow \underset{v\in VG}{\sum }R_{v\left(x_{t}+1,y_{t}\right)},v\times \epsilon \_t$
15:       
$T_{y}\leftarrow \underset{v\in VG}{\sum }R_{v\left(x_{t},y_{t}+1\right)},v\times \epsilon \_t$16:      **If** $T_{x}\leq T_{y}$ **then**17:        $VG\leftarrow v\left(x_{t}+1,y_{t}\right)$
18:         $x_{t}\leftarrow x_{t}+1$.

19:      **Else**

20:         
$VG\leftarrow v\left(x_{t},y_{t}+1\right)$

21:         $y_{t}\leftarrow y_{t}+1$

22:      **End if**

23:    **Else if**
 $dist_{x}<dist_{y}$
**then**
24:      
$VG\leftarrow v(x_{t}+1,y_{t})$
25:      $x_{t}\leftarrow x_{t}+1$
26:    **Else**27:      $VG\leftarrow v\left(x_{t},y_{t}+1\right)$
28:       
$y_{t}\leftarrow y_{t}+1$29:    **End if**30:   **End if**31: **End while**


### 3.2. Double Particle Swarm Based Thermal-Aware Mapping Algorithm

3D on-chip network task mapping is the process of finally mapping a DAG task graph to an NoC resource node. Based on the previous division of NoC resource nodes, the task nodes that are to be mapped and the NoC resource nodes waiting to be mapped have been identified at this point. This section will describe the use of the particle swarm algorithm to map tasks to a NoC. The particle swarm algorithm is a type of swarm intelligence algorithm, which has the characteristics of simple structure and fast convergence for both linear and discrete solution spaces. Each particle in the swarm algorithm represents a solution, which is influenced by the self-confidence value of the particle itself, the optimal solution in each round, and the global optimal solution during the iterative process. However, studies have shown that the particle swarm algorithm is prone to fall into local optimal solutions. Therefore, an improved discrete particle swarm algorithm is proposed in this paper. For the common discrete PSO algorithm, if the task graph has N nodes, we define the structure of particles as a tuple with N elements $p_{k}^{i}=\langle a_{k,1}^{i},a_{k,2}^{i},\ldots ,a_{k,N}^{i}\rangle$, where $p_{k}^{i}$ denotes the *i*-th particle in the *k*-th iteration, and $a_{k,n}^{i}$ denotes the *n*-th particle in $p_{k}^{i}$ task graph node corresponding to the n-th resource node. We define $p_{k}^{i}\to p_{k+1}^{i}$ as the process of particle update, which denotes the exchange sequence of the minimum number of exchanges required to go from particle $p_{k}^{i}$ to particle. For example, if $p_{k}^{i}=\langle 3,1,2,4,5\rangle$ and $p_{k+1}^{i}=\langle 3,2,4,1,5\rangle$, then $p_{k}^{i}\to p_{k+1}^{i}=\langle swap\left(2,3\right),swap\left(3,4\right)\rangle$. We define $p_{k}^{best}$ as the local optimal solution, i.e., the solution with the minimum value of the loss function obtained immediately by the particle swarm algorithm at the k-th round; we define $g_{k}^{best}$ as the global optimal solution, i.e., the optimal solution of the particle swarm algorithm during the iteration. According to the traditional PSO algorithm, the iterative formula for the particle swarm is obtained as follows.
(14)$$p_{k+1}^{i}=\left(\omega_{1}\times I+\omega_{2}\times \left(p_{k}^{i}\to p_{k}^{best}\right)+\omega_{3}\times \left(p_{k}^{i}\to g_{k}^{best}\right)\right)\cdot p_{k}^{i}$$
where $\omega_{1}$ is the inertia factor, defining $I=\langle swap\left(1,1\right),\;swap\left(2,2\right),\;\ldots ,\;swap\left(n,n\right)\rangle$, describing the description of the particle $p_{k}^{i}$ keeping the previous state unchanged in the iteration. In Equation (14), $\omega_{2},\omega_{3}$ is constant, which is ranged between 0–1, and $\omega_{2}\times \left(p_{k}^{i}\to p_{k}^{best}\right)$ represents the probability of each swap operation taking place in the swap sequence $p_{k}^{i}\to p_{k}^{best}$.

In this paper, we proposed a double particle swarm algorithm, the aim of which is to obtain an optimal solution with the fastest running time. In order to prevent the particle swarm algorithm from easily falling into the local optimal solution, the number of particle populations increased to 2. One of the particle populations introduces the property of cross-variation in the genetic algorithm. Another particle population introduces the feature of neighborhood search to increase the local optimization ability. When the two populations converge to the global optimal solution, they also converge to the optimal solution of the other population to avoid falling into the local optimal solution. Therefore, the number of particles that undergo crossover variation and the number of rounds in which the value of the loss function remains constant determine the convergence time of the algorithm. Thus, it achieves controllability in the algorithm execution time and accuracy. When converging, the inertia factor of the PSO is constant, which leads to premature falling into the local optimal solution. The dynamic transformation of the inertia factor is introduced for both particle populations in DPSO to control the speed of convergence to the global optimal solution.

#### 3.2.1. Dynamic Transformation of the Inertia Factor

The inertia index $\omega_{1}$ is an important parameter in the particle swarm algorithm, which is itself an important parameter for the regulation of the global search ability and convergence speed. A larger inertia index $\omega_{1}$ can strengthen the global search ability of t algorithm, but can result in the convergence speed becoming slower; conversely, with a smaller inertia index $\omega_{1}$, the algorithm has poor global search ability, but stronger local search ability and faster convergence speed. Therefore, our goal is for the particle swarm algorithm to have a stronger global search capability in the early stage, and thus be able to cover a larger range in the solution space; in the later stage a rapid reduction in the inertia index $\omega_{1}$ is required in order to strengthen the local search capability.

A more common way to adjust the weights dynamically is to apply a linear weight decreasing strategy, in which the weight is gradually decreased to a smaller value from the early stage of the algorithm. In this paper, an adaptive exponential dynamic transformation weight optimization formulation as in Equation (15) is incorporated. The algorithm ensures that the particle swarm algorithm has a high inertia index in the early stage, which improves the global search ability of the algorithm in this stage; while in the later stage, the inertia index decreases rapidly, which accelerates the convergence speed of the algorithm and the local search ability. The inertia weights are set to the maximum value in the initial stage of the algorithm. During the iteration of the algorithm, the states of all particles are used as input to get the return information, and the rate of inertia index decline is calculated according to the exponential formula. The algorithm uses the evolutionary step of the particles as a feedback parameter and dynamically adjusts the inertia weight strategy according to the particle states, as shown in Equation (15).
(15)$$\left\{\begin{array}{c} \Delta \omega =\frac{\left(\omega_{max}-\omega_{min}\right)}{iter_{max}}\times \left(\frac{iter_{max}-iter}{iter_{max}}\right)\times e^{\frac{\left(iter\right)}{iter_{max}}-d}\\ \omega_{n}=\omega_{n-1}-\Delta \omega \end{array}\right.$$
where Δω denotes the decreasing value of the inertia weight, $\omega_{max},\;\mathrm{and}\;\omega_{min}$ denote the preset maximum inertia weight and minimum inertia weight, respectively. $iter_{max}\mathrm{and}\;iter$ represent the maximum number of iterative rounds and the current number of iterative rounds. d is the environmental feedback factor, and the interval of the value of d is usually around [0, 0.5], which is used to describe the degree of evolution of the particle population in the last iteration. It indicates the feedback property of the particle population to the inertia weight. When the value of d is larger, it means that the evolution of the particles in the last iteration is better, and when the value of d is smaller, the evolution of the particles is worse. If the value of d is 0, it means that the algorithm has converged.

#### 3.2.2. Neighborhood Search

Based on the adaptive exponential dynamic transformation inertia weight optimization formulation proposed in Section 3.2.1, the convergence ability and local search ability of the algorithm are enhanced significantly in the later stage. However, as a swarm intelligence algorithm, the particles with lower adaptability are still attractive to the particles with higher adaptability, and the algorithm rapidly increases the convergence speed in the late stage, which thus increases the probability of avoiding falling into the local optimal solution while improving the local search ability. In this study, to the particle swarm algorithm based on the introduction of adaptive exponential dynamic transformation weight optimization, the concept of neighborhood search is added to search for more optimal solutions in its neighborhood when the particle diversity is less than a set threshold, and the concept of the taboo table is introduced in order to prevent repeated searches. The local search capability of the particle swarm algorithm is increased by masking the particles that have fallen into local optimal solutions and searching for more optimal solutions in their neighborhoods.

#### 3.2.3. Gene Cross-Mutation

Drawing on the idea of a genetic algorithm for avoiding falling into local optimal solutions, in this paper the genetic variation property is introduced into the PSO algorithm. Because the PSO algorithm requires the exploration of a wider solution space upfront, particles must be able to migrate from the solution space more flexibly. We define the structure of particles as a tuple $p_{k}^{i}=\langle a_{k,1}^{i},a_{k,2}^{i},\ldots ,a_{k,N}^{i}\rangle$ with N elements, where $p_{k}^{i}$ denotes the *i*-th particle in the *k*-th iteration, and $a_{k,n}^{i}$ denotes the task graph node corresponding to the *n*-th resource node in $p_{k}^{i}$. We define $p_{k}^{ic}=crossover\left(p_{k}^{i},rand\left(1,n\right),rand\left(1,n\right)\right)$ as the mutation process of particle $p_{k}^{i}$, indicating that the generation of $p_{k}^{ic}$ occurs by the swapping of $p_{k}^{i}$ particles in two random dimensions. The mutation process occurs randomly, which avoids the problem of slowing down the convergence of the particle swarm algorithm after the occurrence of cross mutations and better extends the optimal solution in the early stage of the algorithm. This section uses the environmental feedback factor d, as previously mentioned, as an input to calculate the probability, as shown in Equation (16).
(16)$$\epsilon =\frac{iter_{max}-iter}{iter_{max}}\times d$$

The algorithm requires a larger search range of solution space in the early stage, while in the later stage, it must avoid affecting the convergence effect and local search effect, and the probability decreases with the increase of rounds. However, when the evolution of particles becomes slower, a larger variation probability is needed in order to extend the exploration of the solution space, so the addition of environmental feedback factors can appropriately adjust the search range of the algorithm for the solution space, in order to obtain the optimal solution.

#### 3.2.4. Double Particle Swarm Optimization

To combine the properties proposed above, a double particle swarm optimization algorithm based on cross-variance and neighborhood search is proposed in this section, which combines the strong global search capability of cross-variance and the strong local search capability of neighborhood search as well as the previously mentioned adaptive exponential dynamic inertia factor in order to enhance the integrated capability of the algorithm. Due to the parallel computation of the double particle swarm, a mutual disturbance term is added to Equation (14), as shown in Equation (17).
(17)$$p_{k+1}^{i}=\left(\omega_{1}\times I+\omega_{2}\times \left(p_{k}^{i}\to p_{k}^{best}\right)+\omega_{3}\times \left(p_{k}^{i}\to g_{k}^{best}\right)\right)\cdot p_{k}^{i}+\omega_{4}\times \left(p_{k}^{i}\to q_{k}^{best}\right)$$

$q_{k}^{best}$ is the optimal solution in the double particle swarm, which enhances the linkage of the double particle swarms and further improves the global search capability and local search capability of the algorithm. The algorithm flow is as follows.

The process of DPSO is shown in Algorithm 2.

Initialization: the velocity and position of the particles, based on the initialized position information, are substituted into Equations (9) and (12) to calculate the initial fitness value (lines 1–2). The forbidden table is initialized for the neighborhood search (line 3).The particle population that joins the neighborhood search feature is defined as $pop_{1},$ the list of fitness values as $fitness_{1}$, the particle population that joins the genetic variation feature as $pop_{2}$, and the list of fitness values as $fitness_{2}$ (lines 5–7).Particle evolution: the inertia factor of particles is updated according to Equation (15) and the particle position is updated according to Equation (17) (lines 8–11).The cross-variation state of particles in $pop_{2}$ is evaluated according to Equation (16) (lines 12–17).The diversity of particles in $pop_{1}$ is evaluated, and if the diversity value is less than the predetermined threshold, $g_{1,k}^{best}$ is added to the forbidden table and $g_{1,k}^{best}$ is reset to the suboptimal solution of fitness in $pop_{1}$ (lines 18–23). Step 2 is repeated.

**Algorithm 2**: DPSO
**Inputs:**
$NoC\left(U,L\right)$: NoC topology graph

VG: Group of resource node
X,Y,Z: The number of NoC resource nodes in the three dimensions
$v\left(x,y,z\right)$: Resource node with coordinates x, y, and z
Population: Maximum particle number ${iter}_{m},ax:maximum\;number\;of\;iterations$
**Output: **$MAPPING_{TG}^{VG}$: Mapping relation of TG to VG1: Initializes particle positions and velocities in a random range: $pop_{1}\left[0\right]$, $pop_{2}\left[0\right]$
2: Calculates the fitness of each particle: $fitness_{1}\left[0\right]$, $fitness_{2}\left[0\right]$
3: Initializes forbidden table: ForbiddenTable
4: **For**
$iter<iter_{max}$
**do**5:   updates pop fitness with Equation (9) and Equation (12):6:    $fitness_{1}\left[iter\right]\leftarrow fitness_{1}\left[iter-1\right]$
7:    $fitness_{2}\left[iter\right]\leftarrow fitness_{2}\left[iter-1\right]$
8:   updates inertia factor according to Equation (15)9:      updates particle position according to Equation (17)10:      $pop_{1}\left[iter\right]\leftarrow pop_{1}\left[iter-1\right]$
11:      $pop_{2}\left[iter\right]\leftarrow pop_{2}\left[iter-1\right]$
12:      updates mutation probability13:     **For**
$pop\in pop2\left[iter\right]$
**do**14:       $If\;\mathrm{Equation}\;\left(16\right)\;then$
15:          pop←crossover(pop,random(),random())
16:       **End if**
17:    **End for**

18:    sorts $pop_{1}\left[iter\right]$ according to $fitness_{1}\left[iter\right]$

19:    **If**
$particles_{diversity}<threshold$
**then**

20:        $ForbiddenList\leftarrow pop_{1}\left[iter\right]\left[0\right]$

21:        $pop_{2}\left[iter\right]\left[0\right]$ searches neighborhood, updates if there is a better solution

22:        $pbest\leftarrow pop_{1}\left[iter\right]\left[1\right]$

23:    **End if**
24: **End of**

## 4. Simulation and Results

### 4.1. Experience Environment

This study used the Hotspot-integrated, event-driven C++ 3D NoC simulator. The simulator models the packet communication delays at the clock cycle level. The bandwidth ratio between the horizontal and vertical directions is 100:10. The environment configuration is shown in Table 1.

In this paper, we consider the four baseline algorithms to evaluate the performance of each one, which are simulated annealing algorithm (SA), genetic algorithm (GA), particle swarm optimization algorithm (PSO) and KBMA [15]. and make a comparative analysis of their experimental results.

### 4.2. Experimental Task Graph

In this experiment we used the classical task graphs MWD, VOPD, and DVOPD of NoC as shown in Figure 3, and random task graphs generated by Task Graphs For Free (TGFF) [21].

### 4.3. Performance Comparison

#### 4.3.1. Experimental Results of the Traditional NoC Task Map

Shown in Figure 4, Figure 5 and Figure 6, the performance with different mapping algorithms at different classical task graphs is undertaken with 3D NoC scale set to 4×4×4. Communication delay is reduced by 23.5% on average, the throughput is improved by 8.08% and the energy consumption is reduced by 18.0% in DVOPD, VOPD and MWD task graphs. From the point of view of data transmission, the area division algorithm calculates the thermal resistance and the Manhattan distance between nodes. Due to the small number of nodes in the traditional DAG graph, the same application is divided into a relatively concentrated area. The data transmission passes through a smaller number of nodes and links. The DPSO algorithm can improve the overall performance.

#### 4.3.2. Results of Randomized Task Graphs

Because the number of task graph nodes and NoC are not in one-to-one correspondence, this paper uses an adaptive region partitioning algorithm to partition the 3D NoC and then maps the task graph to task nodes in the 3D NoC according to the partitioning result. As shown in Figure 7, Figure 8 and Figure 9, G64, G48, G32, G24, G16, and G10 are randomly generated task graphs with the number of child nodes of 64, 48, 32, 24, 16 and 10, respectively. It is evident that for the larger task graph G64, the maximum improvement in the average communication delay, throughput rate, and energy loss is 19.48%, 12.48%, and 26.99, respectively, while the context of smaller task graph sizes, such as the task graph with task size 10, has a worse effect.

It is evident that the performance of the DPSO algorithm proposed in this paper improves with the increase of the task graph size in the scenario of 3D NoC scale of 4 × 4 × 4. This indicates that the proposed double particle swarm algorithm has better global search capability in a larger solution space, and that the larger task graph size also allows the region division strategy to make better use of TSV channels and reduce the information communication between layers. When the size of the task graph is smaller than 24 task nodes, the solution space is small, the global search ability and local search ability of the DPSO algorithm cannot be fully reflected, and the effect is close to that of the traditional heuristic algorithm; when the size of the task graph is larger than 24 task nodes, the local search ability and global search ability can be reflected, and the mapping effect is greatly improved. The traditional PSO algorithm is the earliest to fall into the optimal solution, so the overall effect is the worst. From the perspective of data transmission, applications are divided into a relatively concentrated area. When the number of nodes in the task graph is less than 48, there are less NoC nodes and links that data packets pass through. Therefore, this algorithm can achieve better results (lower delay, higher throughput, and lower heat loss) than other algorithms. When the number of nodes in the task graph is greater than 48, the effect of the region division algorithm is weakened. Because the number of nodes in the divided NoC area is close to the number of nodes in the NoC, the best result at this time depends more on the best convergence of the DPSO algorithm.

## 5. Conclusions

In this paper, we first propose an adaptive 3D NoC module partitioning algorithm. The node with the lowest thermal resistance, the node with the lowest delay, and node with the lowest throughput in the neighborhood are searched, starting with the node with the lowest thermal resistance. The DPSO algorithm improves search capability in the early stage based on the PSO algorithm, slows down the convergence of particles in the later stage in order to reduce the probability of falling into local optimal solutions, and adds a neighborhood search function to search for more optimal solutions in the solution space near the optimal and suboptimal solutions. The experiments show that the average delay is reduced by up to 32.4%, the throughput is improved by up to 35.5%, and energy loss is reduced by up to 31.3% compared with the existing algorithm based on a 3D NoC scale of
4×4×4.

## Figures and Tables

**Figure 1 micromachines-14-00628-f001:**
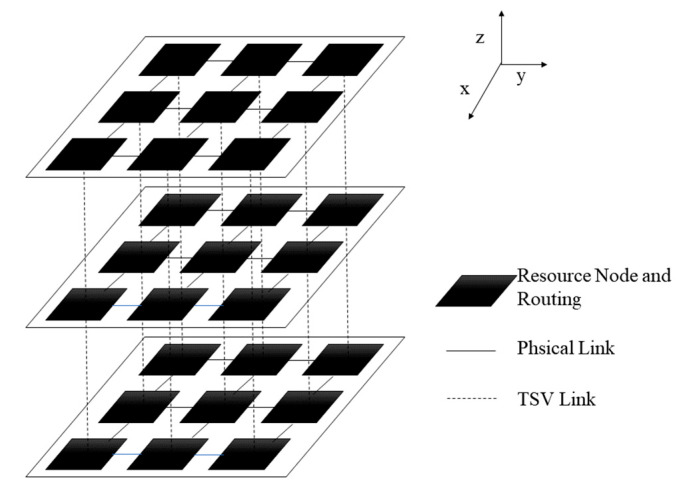
3D NoC architecture graph.

**Figure 2 micromachines-14-00628-f002:**
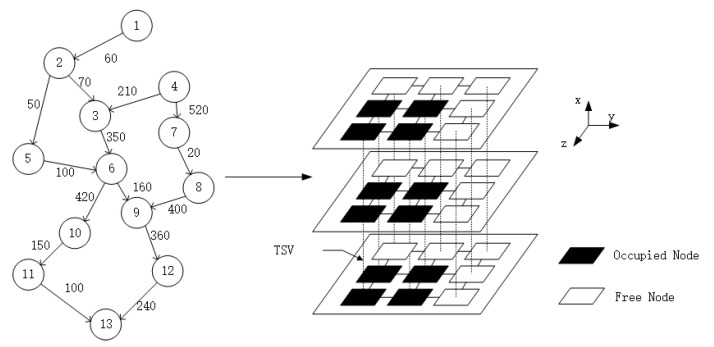
Example map of region division.

**Figure 3 micromachines-14-00628-f003:**
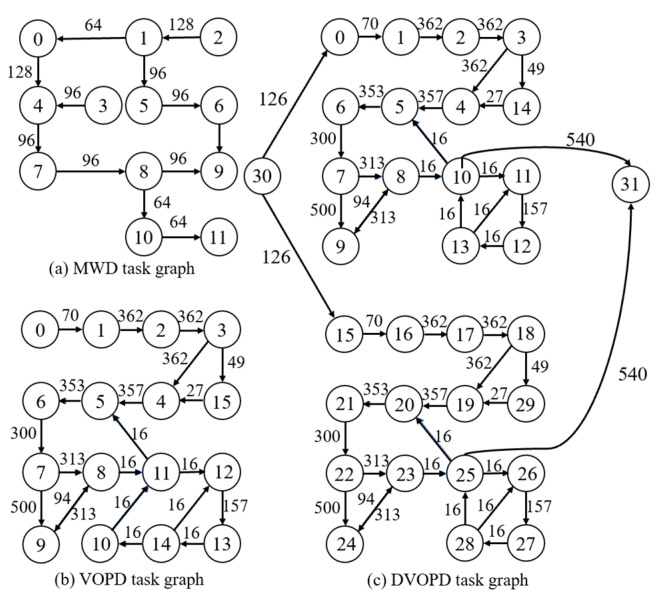
Classic Task Graph.

**Figure 4 micromachines-14-00628-f004:**
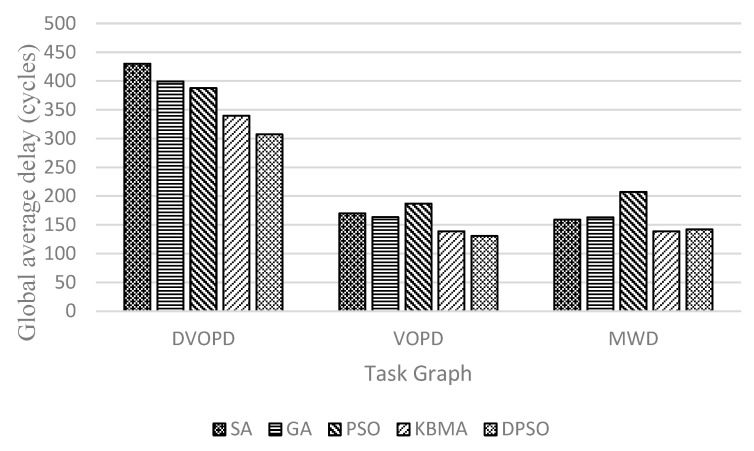
Communication delay at DVOPD, VOPD and MWD on 4 × 4 × 4 3D mesh structure.

**Figure 5 micromachines-14-00628-f005:**
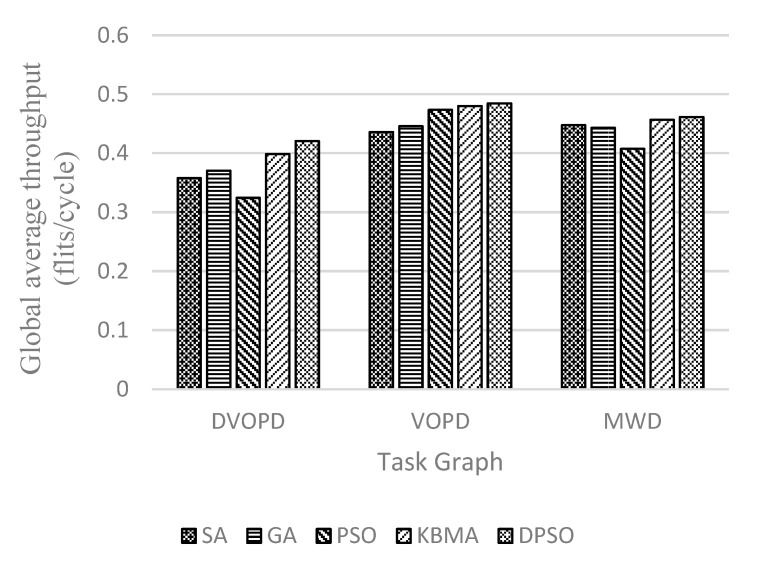
Throughput rate at DVOPD, VOPD and MWD on 4 × 4 × 4 3D mesh structure.

**Figure 6 micromachines-14-00628-f006:**
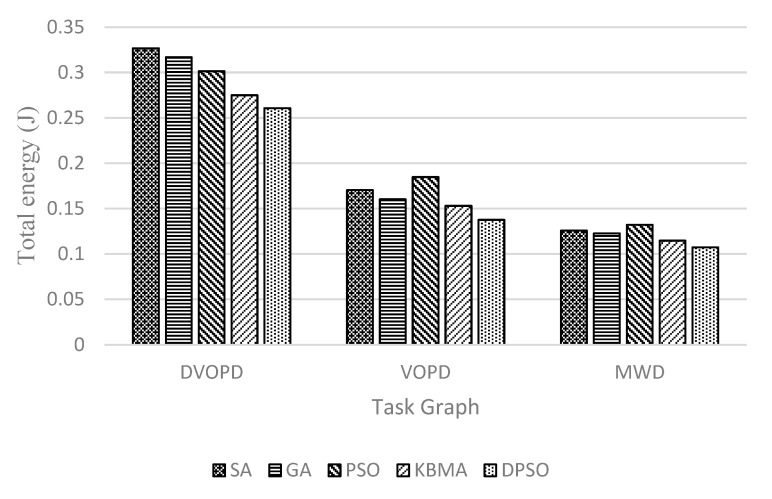
Heat loss at DVOPD, VOPD and MWD on 4 × 4 × 4 3D mesh structure.

**Figure 7 micromachines-14-00628-f007:**
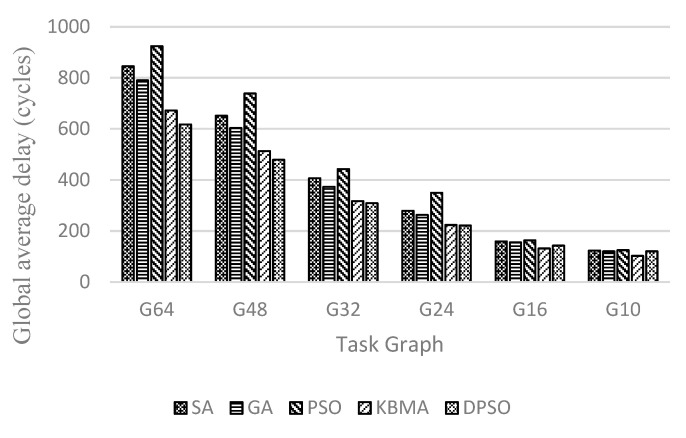
Communication delay at different sizes of random task graphs on 4 × 4 × 4 3D mesh structure.

**Figure 8 micromachines-14-00628-f008:**
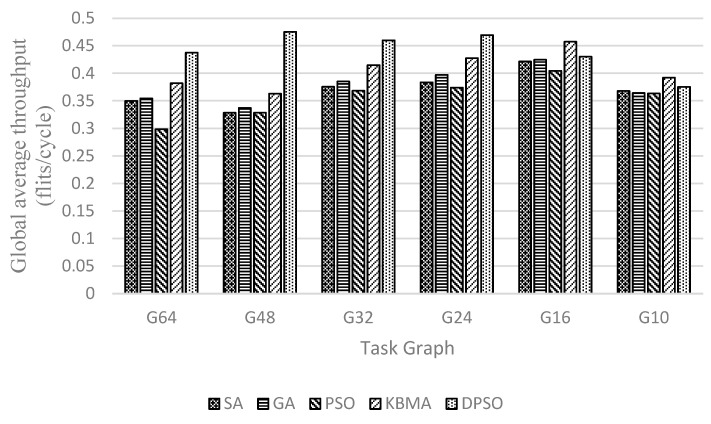
Throughput at different sizes of random task graphs on 4 × 4 × 4 3D mesh structure.

**Figure 9 micromachines-14-00628-f009:**
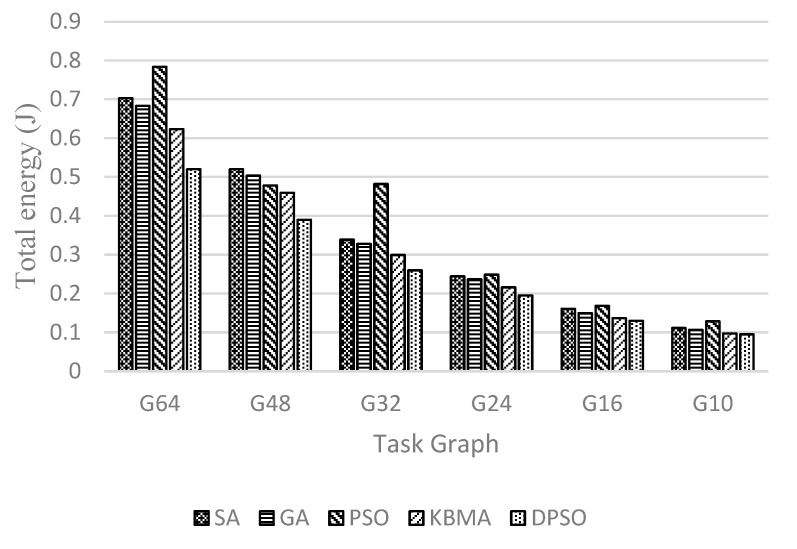
Energy loss at different sizes of random task graphs on 4 × 4 × 4 3D mesh structure.

**Table 1 micromachines-14-00628-t001:** The simulation parameters.

Network parameters	
Flit size	128 bits
Latency Router	2 cycles, link 1 cycle
Buffer depth	4 flits
Routing algorithm	XYZ routing
Baseline topology	4×4×4
Configuration of Many-core Simulator	
Core Architecture	Intel Pentium 4
Baseline Frequency	1.9 GHz
Hotspot Parameters	
Die size [mm]	0.5 × 0.5
Specific heat capacity [J/(m^3^ × K)]	1.75 × 10^6^
Resistivity [(m − K)/W]	0.01

## Data Availability

Not applicable.

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
