# Peer review of "Hybrid Optimization Algorithm Based on Double Particle Swarm in 3D NoC Mapping"

_micromachines, 2023, doi:10.3390/mi14030628_

Round 1
Reviewer 1 Report
The paper proposes an evolutionary algorithm for mapping task to NoCs.
The main drawback which should be improved is evaluation part. Currently the proposed algorithm has been compared with three basic algorithms. It will be better to compare with some algorithm published in recent years.
References from recent years should be added in the introduction and simulation results parts. The newest reference is for 2021 (only one reference) and 2019 (two references) which is insufficient.
Author Response
Response to Reviewer 1 Comments
Thank you for your feedback and suggestions. We have incorporated all comments and suggestions into the revised version of the paper. The following is a summary of the revision we conducted.
Point 1: The main drawback which should be improved is evaluation part. Currently the proposed algorithm has been compared with three basic algorithms. It will be better to compare with some algorithm published in recent years.
Response 1: Thank you very much for raising this issue. I did some experiments to complement a new baseline algorithm called KBMA which was proposed in 2019.
Point 2: References from recent years should be added in the introduction and simulation results parts. The newest reference is for 2021 (only one reference) and 2019 (two references) which is insufficient.
Response 1: Thanks for the comment. I added some references in the introduction.

Reviewer 2 Report
In this paper, the authors presented a 3D network-on-chip partitioning algorithm. However, the contribution is rather trivial and
and has less theoretical depth. The composite heuristic simulation algorithm arisen has been proven to
have lower accuracy in learning simulation. The writing is poor. The novelty is not enough since comparison with existing results
is missing and the related important references are not cited.
Besides, English grammar, spelling,
and sentence structure are too poor to be understood. Too many randomly and unnecessary brevity terms decrease the readability. The space before and after the punctuations are chaotic. The tenses are not unified in a same paragraph. In P1 L7,two "Correspondence"s are listed. Two many uppercase letters should be lowercase ones. The cited authors should be proposed only surnames but not first names.
Two many "where"s should not start a new paragraph. The format of function and variable fonts are confusing. The titles in references should not be written with uppercase letters for every word.
The following related complex nonlinear system algorithm analysis references should be cited to
highlight the motivation.
[1] Low-complexity tracking control of strict-feedback systems with unknown control directions, IEEE Transactions on Automatic Control, 2019, 64(12): 5175-5182.
[2] Image enhancement based on rough set and fractional order differentiator, Fractal Fract, 6(4):214, 2020.
[3] Analysis and prediction of COVID-19 epidemic in South Africa, ISA Transactions,124 (2022) 182-190.
Author Response
Response to Reviewer 2 Comments
Thank you for your feedback and suggestions. We have incorporated all comments and suggestions into the revised version of the paper. The following is a summary of the revision we conducted.
Point 1:
The contribution is rather trivial and has less theoretical depth. The composite heuristic simulation algorithm arisen has been proven to have lower accuracy in learning simulation. The writing is poor. The novelty is not enough since comparison with existing results is missing and the related important references are not cited.
Response 1: Thank you very much for raising this issue. This paper proposes a double particle swarm optimization algorithm. One of the particle populations introduce cross-mutation to expand the search range of the solution space. Another particle population introduce tabu search to improve local search capabilities. When the two populations update their vectors of positions to the global optimal solution and the local optimal solution, they also update to the optimal solution position of the other population. And a dynamic inertia factor is proposed to control the convergence speed.
The paper has been checked by native speaker.
I have added some experiments results to complement a new baseline algorithm called KBMA which was proposed in 2019.
Point 2: Besides, English grammar, spelling, and sentence structure are too poor to be understood. Too many randomly and unnecessary brevity terms decrease the readability. The space before and after the punctuations are chaotic. The tenses are not unified in a same paragraph. In P1 L7,two "Correspondence"s are listed. Two many uppercase letters should be lowercase ones. The cited authors should be proposed only surnames but not first names.
Two many "where"s should not start a new paragraph. The format of function and variable fonts are confusing. The titles in references should not be written with uppercase letters for every word.
Response 2: We are very sorry for the mistakes in this manuscript and inconvenience they caused in your reading. The manuscript has been thoroughly revised and edited by a native speaker, so we hope it can meet the journal's standard. Thanks so much for your useful comments.
Point 3: The following related complex nonlinear system algorithm analysis references should be cited to highlight the motivation.
[1] Low-complexity tracking control of strict-feedback systems with unknown control directions, IEEE Transactions on Automatic Control, 2019, 64(12): 5175-5182.
[2] Image enhancement based on rough set and fractional order differentiator, Fractal Fract, 6(4):214, 2020.
[3] Analysis and prediction of COVID-19 epidemic in South Africa, ISA Transactions,124 (2022) 182-190.
Response 3: Thank for your comments. I have cited these references to highlight my motivation.

Reviewer 3 Report
This paper proposes an extension of the PSO algorithm to assign nodes in a 3D NoC architecture with the aim to reduce energy used.
In the introduction, you seem to imply that you propose a 3D NoC on a 2D SoC system. If you are already using through Silicon vias for the NoC, why don’t you also use them to create a fully 3D system with the 3D NoC integrated in it? In later sections (e.g., the beginning of section 3), you indeed hint at a fully 3D system with more cores than a 2D system. You should explain well what kind of system you are targeting.
There are many language errors in the paper that need to be addressed. Many sentences are ill-formed (missing verbs, sudden changes in meaning, etc.). This makes your paper very hard to read and to understand.
Your paper lacks enough structure to be fully comprehensible. You seem to jump from one discussion to another one and never is it really clear how exactly the current discussion is contributing to the overall goal (which is also not clear as you seem to hint at wanting to reduce mostly power consumption but the algorithms do not seem to focus on this alone. You should explain your contributions much better and give an overview of your contributions in the introduction.
On page 7 you seem to describe an algorithm but you did not discuss what the itemized list actually is nor indicate that you are going to describe an algorithm. What is the purpose of it? Also, what are "longitudinal neighboring nodes”
You do not refer to figure 1 nor to algorithm 1 in the text.
You use the particle swarm optimization algorithm but never really motivate why that would be a good choice. On the contrary, because studies have shown that the particle swarm algorithm is prone to fall into local optimal solutions, you had to propose an improved discrete particle swarm algorithm. Please motivate your choices better.
Also, it is not clear at all how the Double PSO algorithm differs from PSO and why it is called Double PSO.
You claim you “introduce” the genetic variation property into the PSO algorithm. However, genetic variation is inherently present in PSO so it is not introduced by you.
Again, you do not refer to Algorithm 2 at all.
In section 4.2.1, you say you can verify the effectiveness of the DPSO algorithm but you do not refer to results to back up that claim. The reader has to read through the entire paragraph until at the end a reference to figures 3-5 is mentioned.
The results in figures 3-5 do not show very good results for PSO. Only for DVOPD your PSO algorithm is better than the others. However, DPSO is better for all of them Why did you put this as the first bar in the figures? It would seem more logical to put that result as the last bar as this is your proposal that you compare to previous results.
Your results are for small NoCs with only 4x4x4=64 nodes. You claim that the PSO results would be better for larger sizes but why did you not show that in your paper? You could gradually increase the size from 3x3x3 to 6x6x6 e.g. and show the scaling behavior.
Details
You are wrongly referring to “Section 1” as “Chapter 1”.
In the beginning of section 3.2.2 you refer to “the previous paper”. Which paper is that?
You should leave some vertical space between the text and the algorithms.
Captions should be on the same page as the figure (fig. 3 and fig. 8).
Author Response
Response to Reviewer 3 Comments
Thank you for your feedback and suggestions. We have incorporated all comments and suggestions into the revised version of the paper. The following is a summary of the revision we conducted.
Point 1:
In the introduction, you seem to imply that you propose a 3D NoC on a 2D SoC system. If you are already using through Silicon vias for the NoC, why don’t you also use them to create a fully 3D system with the 3D NoC integrated in it? In later sections (e.g., the beginning of section 3), you indeed hint at a fully 3D system with more cores than a 2D system. You should explain well what kind of system you are targeting.
Response 1: Thank you very much for raising this issue. We have added the explanation of 3D NoC architecture in section 2.2.
Point 2: There are many language errors in the paper that need to be addressed. Many sentences are ill-formed (missing verbs, sudden changes in meaning, etc.). This makes your paper very hard to read and to understand.
Response 2: Thanks for the comment. The paper has been checked by native speaker.
Point 3: Your paper lacks enough structure to be fully comprehensible. You seem to jump from one discussion to another one and never is it really clear how exactly the current discussion is contributing to the overall goal (which is also not clear as you seem to hint at wanting to reduce mostly power consumption but the algorithms do not seem to focus on this alone. You should explain your contributions much better and give an overview of your contributions in the introduction.
Response 3: We have added discussion of contributions to improve the readability of the paper and summarizes the contributions in the introduction of the paper.
Point 4: On page 7 you seem to describe an algorithm but you did not discuss what the itemized list actually is nor indicate that you are going to describe an algorithm. What is the purpose of it? Also, what are "longitudinal neighboring nodes”
Response 4: We are very sorry for the mistake. I have discussed algorithm 1 in detailed.
The purpose of Algorithm 1 is to divide some 3D NoC resource nodes, so that the thermal influence and Manhattan distance between these nodes are minimized, so as to reduce the solution space of DPSO and increase the convergence speed.
Point 5: You do not refer to figure 1 nor to algorithm 1 in the text.
Response 5: Thank you so much for your careful check. According to the revised content, we have refer to figure 1 and algorithm 1.
Point 6: Also, it is not clear at all how the Double PSO algorithm differs from PSO and why it is called Double PSO.
Response 6: Thank you for pointing out this problem in manuscript. I have added related discussion in lines 355-369, section 3.2.
In order to prevent the particle swarm algorithm from easily falling into the local optimal solution, the number of particle populations increased to 2. One of the particle populations introduce the property of cross-variation in the genetic algorithm. Another particle populations introduces the feature of Neighborhood search to increase the local optimization ability. When the two populations converge to the global optimal solution, they also converge to the optimal solution of the other population to avoid falling into the local optimal solution.
Point 7: You claim you “introduce” the genetic variation property into the PSO algorithm. However, genetic variation is inherently present in PSO so it is not introduced by you.
Response 7: Thank for your comments. We believe that our main contribution is that two populations of particles, while converging towards the global optimal solution, the local optimal solution, and also converge towards the optimal solution of the other particle population. A dynamic inertia factor is also proposed to control the convergence speed.
Point 8: Again, you do not refer to Algorithm 2 at all.
Response 8: We are very sorry for the mistake. We have refered to Algorithm 2 in the paper.
Point 9: In section 4.2.1, you say you can verify the effectiveness of the DPSO algorithm but you do not refer to results to back up that claim. The reader has to read through the entire paragraph until at the end a reference to figures 3-5 is mentioned.
Response 9: We feel sorry for the inconvenience brought to the reviewer. We moved the reference to the front of the paragraph, so that the experimental data can be seen for the convenience of the reader.
Point 10: The results in figures 3-5 do not show very good results for PSO. Only for DVOPD your PSO algorithm is better than the others. However, DPSO is better for all of them Why did you put this as the first bar in the figures? It would seem more logical to put that result as the last bar as this is your proposal that you compare to previous results.
Response 10: We feel sorry for the inconvenience brought to the reviewer. We adjust the DPSO algorithm to the last bar in the figures. In addition, We have added the experimental results of baseline algorithm, called KBMA, as a comparison experiment.
Point 11: Your results are for small NoCs with only 4x4x4=64 nodes. You claim that the PSO results would be better for larger sizes but why did you not show that in your paper? You could gradually increase the size from 3x3x3 to 6x6x6 e.g. and show the scaling behavior.
Response 11: We totally understand the reviewer's concern. In our experiments, the size of the 3D NoC is 4*4*4 constant, and the number of nodes in the task graph is variable. What we claim is that the DPSO algorithm has better convergence results for the larger task graph.
We are very grateful for your comments, we will continue to discuss in follow-up research according to your guidance, and compare with previous research, continue to contribute

Round 2
Reviewer 1 Report
It seems necessary author add some words in the beginning of section 4.2.1 and also a briefing sentence between 4.2 and 4.2.1.
Reviewer 2 Report
The manuscript has been modified according to my comments. Minor writing revision is needed including the grammar and format. It is acceptable now.
Reviewer 3 Report
The authors have taken most of my comments into consideration